# Pharmaceutical Care in Sports

**DOI:** 10.3390/pharmacy8040218

**Published:** 2020-11-16

**Authors:** José Henrique Gialongo Gonçales Bomfim

**Affiliations:** Grupo Técnico de Suplementos Alimentares, Conselho Federal de Farmácia–CFF–Brasil, Brasília DF CEP 71635-615, Brazil; drhenrique@msn.com; Tel.: +55-16-988041154

**Keywords:** sports pharmacy, drugs in sports, pharmaceutical care, pharmacotherapeutic follow up

## Abstract

Pharmaceutical care in sports is a new field of work to clinical pharmacists, focused on promoting pharmacotherapeutic follow up and clinical services to athletes, physical activity practitioners and enthusiasts of any sports modality. A broad range of pharmaceuticals, dietary supplements and herbal drugs have been used historically as performance promoters, doping or ergogenic aids. In this context, the role of pharmacists in prevent adverse events, drug interactions or any drug related problems, as doping issues, was described. Its actions can be important to contribute with a multi professional clinical health team, leading athletes to use these resources in a rational way, promoting and optimizing the therapeutic when its necessary.

## 1. Introduction

There are some important issues surrounding the drug utilization by professional athletes and exercise enthusiasts. Despite the fact that medicines are used as important tools in the process of injury recovery by athletes, a large number of pharmaceuticals have been used with different objectives and purposes, many times illegally, with the aim to achieve aesthetic and ergogenic results. Thus, pharmacists can play an important role promoting a pharmacotherapy follow up and providing pharmaceutical care, focused to assess whether the drug use can be harmful and to identifying and preventing adverse effects, drug interactions and any other drug related problems. Therefore, for this purpose, it is important to review the concept of pharmaceutical care and its goals.

Pharmaceutical care is the integrated action of the pharmacist with the health team, centered on the user, for promotion, protection and recovery of health and prevention of diseases related to the use of medicines. In Brazil, the concept of pharmaceutical care is used to define a broad set of clinical actions promoted by pharmacists, and there are two ways that the term is translated: “atenção farmacêutica” or “cuidado farmacêutico” [1]. Both terms are directly related to the pharmacist’s clinical role on improving the patient’s quality of life. These actions aim at health education and the promotion of the rational use of prescription and non-prescription drugs, alternative and complementary therapies, through the services of the pharmaceutical clinic and the technical-pedagogical activities directed at the individual, family, community and health team [2].

Since the publication of “Proposal of the Brazilian Consensus on Pharmaceutical Care”, in 2002, there has been a large growth in publications focusing on the practice of pharmaceutical care, carried out in the most diverse environments and in the results of the clinical actions of the pharmacist, demonstrating the direct impact of these actions in the prevention of problems arising from pharmacotherapy and health promotion, to the evaluation of the results of the pharmaceutical intervention in specific situations and their direct importance in improving the quality of life of the patients [3].

Pereira and Freitas, in 2008, had already demonstrated the perspectives of the practice of pharmaceutical care in the country, pointing out a growth of this clinical movement that would be consolidated, later, with the publication of specific resolutions, directed exclusively at clinical practice (RDC ANVISA 44/09 and RDC CFF 585/13), and finally to pharmaceutical prescription (RDC CFF 586/13), which resulted in a milestone of this movement as a direct trend of the profession and has been opening space for several other areas inserted in clinical practice [4,5,6,7].

In this context, pharmaceutical care is part of the set of clinical actions that involves other health professionals in a multidisciplinary work, with the main focus on the patient, seeking the resolution of situations associated or not with pharmacotherapy, where the clinical pharmacist plays a fundamental role in solutions.

Given this scenario, the clinical pharmacist has the skills and technical abilities to act directly with any drug user, even when these products do not have a direct indication related to a treatment for a specific pathology, as in the case of physical activity practitioners and professional athletes. This group of individuals usually uses medicines, supplements and herbal products with objectives different from those directed from patients who are users of drug therapy.

Therefore, the aim of the present work was to describe the role of the clinical pharmacist and its actions related to the sport practice, in the context of pharmaceutical care, and demonstrate how this specific service can optimize the use of drugs, supplements, and herbal medicines by athletes and physical activity practitioners.

## 2. The Pharmacist and the Sports Practice

Historically, the relationship between the pharmacist and the sports practice is directed to the doping control, focused on the laboratory exams and results of such analyzes or, more recently, on the clinical aspect of monitoring in situations directed to the use of medicines in the recovery process of the athlete.

The clinical aspects of the pharmacist’s actions in sports are explored in several publications that depict the “sports pharmacy” as a new professional scope, a new specialty, or a new acting opportunity. Drug use by athletes, questions related to doping and their prevention and the use of dietary supplements, are presented in articles published by researchers from different countries.

Ambrose (2004) describes the opportunities and obligations of pharmaceutical professionals in relation to doping control, both in the use of drugs and supplements and other compounds that are used to improve performance (ergogeny). The author concludes that pharmacists can play a key role in participating in doping control programs in addition to preventing athletes from inadvertently consuming a substance banned by agencies or federations [8].

This type of athlete counseling service by the pharmacist is explored by Kasashi (2012), Yamaguchi et al. (2013), Nakajima et al. (2018) and Chiang et al. (2018). All articles focus on the issues of orientation and education of athletes in relation to substance use and prevention of doping, through the actions of pharmacists. Kasashi describes that athletes usually seek medical treatment for problems not reported to the medical team in their sports modality. In these situations, the prescription of medicines that may contain substances present in anti-doping lists is common, and this type of problem could be solved with the presence of the pharmacist on the sports team. In summary, all the results demonstrate that athletes and professionals understand the importance of the pharmacist as a member of the sports team, but there is still a need to expand the knowledge related to the use of drugs in sports. [9,10,11,12].

In addition to professionals, the issues surrounding the training of pharmacy students focused on sports are explored by Ambrose (2008) and Shibata (2017). All the articles show that there is a great interest on the part of the students in deepening in the sport pharmacy modality, since this specialty is still little explored in several countries and the opportunities are numerous, within the teams in the doping controls and advice on the use of substances by athletes [13,14].

Hooper et al. (2019), reviewed potential roles for pharmacists in sports medicine. The authors describe sports pharmacy as an emerging specialty, with focus on doping prevention and control, injury prevention and management, and educational opportunities. The main point of this review is based on demonstrating the possible opportunities for pharmacists in this new field of activity, but the clinical aspects of the pharmacist’s role in provide pharmaceutical care are not explored [15].

## 3. Dietary Supplements, Herbals and Drug Use by Athletes

Practitioners of physical activities and athletes frequently use various classes of medications and innumerable food and herbal supplements for the most diverse purposes within sports practice, such as recovery of injuries, performance gain, prevention of adverse events related to the drugs themselves, and even doping, and, often, they do not have adequate follow-up and orientation to the use.

The major problem is that this use may be inadequate in a number of aspects, such as lack of evaluation and need for use, lack of prescription, acquisition of special control drugs through illegal means, use of non-approved substances for use in humans, dosages without scientific evidence of safety, and other practices that may be harmful to the user. Bodybuilding is a historical example of modality where there is indiscriminate and inadequate drug use.

In this modality, the use of medicines can begin as soon as the practitioners initiate the activities in the academies. These places often end up functioning as a point of sale for the products and it is common for this to be propagated in this environment, totally clandestine and without professional participation, evaluation, and necessity [16]. Issues involving the use of drugs in sports practice are under-reported and there is no reliable information on how these products are being used, as well as adverse reactions, drug interactions and/or nutrient—drug interaction, poisoning or even death related issues.

The medicines are obtained without prescription, through a clandestine market and coming from foreign laboratories, not having even a registration with the Brazilian Health Surveillance Agency (ANVISA), and thus there is no way to carry out any kind of tracking of batch problems, factors that involve good manufacturing practices and laboratory quality, adverse drug effects and any other topic related to pharmacovigilance and monitoring of such products [17].

Some local epidemiological studies portray the panorama of the use of medications by bodybuilders quantitatively and describe what products are used, motivation of use, adverse reactions and the socioeconomic profile of this use. These studies are usually performed through questionnaires sent to participants and there may be a bias in the documented responses [18,19,20,21].

There is great taboo and bias in reporting usage and this does not give us access to accurate information about what happens, from the use of amateur bodybuilders to high-performance athletes. In this context, the pharmacist has a fundamental role in the monitoring of these athletes, with the purpose of detecting any situations that involve the abuse of these products.

However, in a few sporting modalities, as in the case of bodybuilding, substance use is “tolerated”. The issues surrounding the use of aesthetic drugs are controversial and generate heated debates in the medical community.

Among the substances used by athletes of this modality, the most popular are the androgenic anabolic steroids (AAS), substances derived from testosterone and that have indication of therapeutic use for various diseases and situations, such as hypogonadism, cachexia, sarcopenia, caloric intake for patients after large surgeries or loss of muscle tissue and other various syndromes (e.g., Turner, Cushing) and androgen replacement due to advanced age and anemic processes, among others [22,23,24,25].

In Brazil, the commercially available AASs are the testosterone esters (e.g., cypionate, propionate, enanthate, decanoate) and chemical derivatives (e.g., oxandrolone, nandrolone, metenolone, drostanolone, and stanozolol). All substances are under control of federal laws. The other testosterone derivatives substances are not usually prescribed or allowed in human beings or are not released in the country. There is no type of study that determines which dosages are safe and which drugs can be used by athletes in bodybuilding in relation to aesthetics and gain of muscle mass, but the practice has been consolidated for decades and AASs enter as a determining factor in the competitions. Several articles and reports from anti-doping agencies mention the abuse of such substances among athletes of various modalities [26,27,28,29].

When asked about the use of anabolic steroids and other drugs, athletes usually omit this fact, due to a great prejudice related to the use of these substances. The information about use, protocols, dosages, and adverse effects, is usually obtained by athletes in a literature not considered scientific, but widely known among practitioners, such as the books “Musculação—Anabolismo total” [30], “Anabolic steroids—a question of muscle” [31] and “Anabolics” [32]. In these books, protocols for the use of AAS, as well as other drugs, are described in various situations, such as pre-competition, recovery, off-season or training. These protocols are also known in the world of bodybuilding as “cycles”. However, there is currently no direct reference that can direct the reader to where such dosages and drug associations have been obtained.

These “cycles” usually last for weeks, with doses between 10 and 100 times higher than those described as therapeutic in the medical literature [33], as well as associations to avoid the appearance of adverse reactions such as gynecomastia, hyperprolactinemia, hepatotoxicity, hypogonadism among others. For each “cycle” there is also a protocol called “post cycle therapy” or PCT, in addition to the “protection protocol”, which involves the use of gonadotrophic coronary hormone (HCG), growth hormone (GH), testosterone esters, selective estrogen receptor modulators (SERMs) as tamoxiphen and clomiphene, aromatase ihninitors as anastrazole and letrozole, vitamins of the B complex and hepatoprotectors (silymarin, glutathione), in order to prevent testicular atrophy, reduction in testosterone levels, hypogonadism, gynecomastia, and hepatopathies [34].

Currently, there are cycles where athletes use drugs known as selective androgen receptor modulators (SARMs), such as ostarine (enobosarm), andarine, testolone, and ligandrol (cardarine and ibutamoren are marketed as SARMs but these substances have different mechanisms). These drugs would be non-hormonal options to the AAS and act as agonists of the androgen receptor (main binding protein of the AAS) and are often used experimentally or with indication for situations such as stress urinary incontinence and cachexia related to a specific type of diseases such as breast cancer and chronic obstructive pulmonary disease [35,36,37]. The use of this class of products has been growing because there is a collective concept, without scientific basis, that they are safer in relation to AAS and do not present adverse reactions, mainly related to androgenization, besides their sale being facilitated by the internet, since many are still being tested or even abandoned by the original laboratories [38].

In addition to AAS and SARMs, there is also a growing use of so-called hormone precursor peptides (thermorelin, sermorelin, hexarelin, and ipamorelin). These products are amino acid fragments, used as substances that increase endogenous production of hormones such as GH and ghrelin, providing direct anabolism and frequently described as safer options for AAS. However, there is no clinical evidence of its use, safety, or even registration of clinical trials on the official clinical trials.gov website (NIH database for insertion of drug studies with humans).

Together with this range of medications, used without a clinical basis for aesthetics, athletes still use protein and vitamin supplements, supplements known as “pre-workout”, which have sympathomimetic agents such as caffeine and ephedrine in their composition, as well as other products for vasodilation, increased cardiac output and decreased adipose tissue, and any other product that has some ergogenic effect, such as insulin and several hormones [39].

The use of supplements and herbal products is stimulated through social media and internet, with athletes and personalities of the “fitness world” as their main promoters. This often means that people who are beginning to practice physical activities already seek such products, without at least conducting an assessment fulfilled by professionals such as nutritionists, physical education professionals and doctors, to evaluate their real need for use, believing that these products can bring immediate results.

Dietary supplements should be used as a tool, a strategy to complement the athlete’s diet or when the professional involved understands that their use is necessary. Most of the time, supplements can provide a small increase in yield in specific modalities. Every year, International Society of Sports Nutrition (ISSN) exercise and sports nutrition review and International Olympic Committe (IOC) consensus are published and bring reviews about supplements used in sport that have a greater degree of evidence in terms of improving the athlete’s performance.

For endurance athletes, the use of energetic supplements (carbohydrates), beta-alanine, sodium bicarbonate, caffeine, and nitrates have an important role due the stronger evidence for its benefits in these modalities, as energy promoters or performance enhancers. Creatine and nitrates have the same level of scientific evidence when used for high-intensity and submaximal exercises, as well as muscular strength and power promoters [40,41].

In the case of herbal products, some are used to enhance muscle strength, body mass, increase mental vigilance, stimulate fat-burning metabolism, and improve performance as ergogenic aids. These propose benefits are attributed by the presence of a wide range of bioactive compounds in plants, such as polyphenols, terpenoids, and alkaloids. Ginseng, guarana, green tea, ginger, *Tribulus terrestris*, *Gingko biloba*, fenugreek, *Salix alba*, and saffron are some common examples of plants that have been used by athletes as an performance enhancers, with some consistent evidence of their benefits [42].

In total, a bodybuilding athlete can continuously use more than 15 different drugs per protocol, in supratherapeutic dosages, with no scientific and/or clinical evidence of their use, in an off-label way in the majority of cases and without prescription thereof.

The most common adverse reactions are related to the androgenic effects of testosterone derivatives, which when administered are converted to dihydrotestosterone (DHT) by the enzyme 5-alpha reductase in some tissues, or into estrogen when converted by aromatase into other tissues. Among to adverse reactions include: alterations in the lipid profile, alterations in the hepatic and renal functions, alterations in the hematocrit. Other common observed reactions are: gynecomastia, acne, alopecia, cardiovascular alterations, aggression, post-use depression, and hypogonadism in the withdrawal of medications [43,44].

When prescribed, the professionals involved are sports doctors and endocrinologists, but there is no description in the literature of safe dosages related to its use with aesthetic purpose, where prescriptions are then based directly on dose empiricism and dosing interval [28].

In other modalities, such as endurance (cycling and running), there is a prevalence of use of stimulants or even of AAS, but those fall under the concept of competitive doping and, therefore, may be of lesser intensity in relation to use in bodybuilding. However, the use of supplements in this modality is more common and associated with a high degree of evidence of benefit for athletes, such as energetics (carbohydrates in several forms), buffers (beta alanine and bicarbonate) and other substances.

Although described in the literature, substance abuse by athletes, especially bodybuilders, needs to be portrayed in detail, in order to understand the issues involved in this practice, which is often clandestine and may be associated with severe health problems.

## 4. Pharmaceutical Care and Clinical Services to Athletes

As described previously, clinical actions linked to pharmaceutical care tend to be most effective when the pharmacotherapeutic follow-up service is applied to users. It is in this type of service that the professional can identify all the problems related to the pharmacotherapy and, with this data, propose means of resolution or even prevention of these.

For this, the professional makes use of clinical methods of follow-up that involve questionnaires of pharmaceutical anamnesis and collection of data described by the patient, prescribers and laboratory data. In this way, the pharmacist will have access to a range of products used, from medicines to herbals and this will allow an adequate evaluation of the therapeutics, making it possible to find possible errors, drug interactions, or medicines in use without real need.

In the case of athletes or physical activity practitioners, the professional can use already established clinical methods such as *SOAP* (subjective, objective, evaluation, and plan) [45], Dáder Method [46], Pharmacotherapy WorkUp [47], and Therapeutic Outcomes Monitoring (TOM) [48], adapted to the format of the specific need for practitioners of sports modalities. All methods consist of anamnesis questionnaires and collection of all data of medicines used, dosages and indication of their use. From this data, the professional can identify if there are problems related to pharmacotherapy and this allows him or her to take actions pertinent to its scope or even promote referral in more complex situations involving the opinion of other professionals.

For the collection and analysis of data on dietary supplements used by athletes, the WEIGHT$ method described by Morris (2015) is an example of a questionnaire that can be used. The method consists of a seven-point protocol, using the acronym WEIGHT$, where each letter means a question or objective, in order to access important points about the use of supplements by athletes (W—What’s the goal?; E—Exercise type?; I—Indicate who recommended supplement; G—Gather personal history and investment; H—How effective is the supplement?; T—The risk of the supplement?; $—Cost-effective?). In this protocol are evaluated the purpose of the use of the supplement; type of sports modality; who indicated or recommended the supplement; medical history data, efficacy of the supplement (based on the body of evidence and degree of recommendation for its use); risks associated with its use and cost-effectiveness [49].

It is important to note that drugs, supplements, and herbal medicines are used several times through self-medication and users only search for information about the products via the Internet or other electronic means, without the assistance of qualified health professionals. On the other hand, many professionals still need better technical training, mainly in relation to the use of supplements and phytotherapy and its direction to the sport.

Howard et al. (2018) evaluated the interest of athletes in receiving advice on the use of supplements through pharmacists and also if the professionals had training and technical support for such activity. The authors concluded that pharmacists were not identified as a primary source of guidance for sports supplements, but athletes would be willing to discuss this topic with experienced pharmacists. In the article, it has been reported that pharmacists did not believe they had knowledge and confidence regarding supplements used by athletes, but they noted enthusiasm on the part of the practitioner in providing counseling. Finally, the authors conclude that counseling in the sports pharmacy could be a viable expansion of services in community pharmacies with adequate education and tools [50].

But some important concerns must be considered when these kind of products are used. With a growing market in the past ten years around the world, the number of brands and options has increased considerably, and the risk of adulteration and lack of quality are important issues to be taken into account by a pharmacist. In this scenario, it is not uncommon to find products with questionable quality and without any evidence on its benefit as performance enhancers.

These products are promoted in market with some claims as “testosterone boosters”, “fat burners”, “performance enhancers”, and “muscle mass gainers” and a lot of publicity given to these products can lead the general public, as well sports enthusiasts, to buy these products believing in the miraculous way to gain potency and performance. The myth of “natural” is often used as a market strategy to sell herbal products and dietary supplements, with the general population still believing that plant-based products are free of toxic effects.

There is a vast amount of literature that describe how herbal products can be contaminated by drugs and other agents, as well as problems involving dosages and other issues that can be harmful to the final user [36,51,52]. Therefore, the pharmacist’s role in this case is essential to avoid unnecessary drug-related problems and to promote a rational use of these products, evaluating dosages and formulations, in order to detect and prevent harmful situations that can arise from the inappropriate use of herbal and dietary supplements.

In addition to pharmacotherapeutic follow-up, it is the pharmacy’s responsibility to provide other clinical services that may contribute to optimizing the therapy of athletes and physical activity practitioners. The administration of medicines, for example, is a permitted and extremely important service, since it allows the user to receive his medication in an appropriate way, which is administered by a professional with training and technical knowledge.

In the academy environment, it is common to report injecting forms without safety criteria and adequate techniques, which can lead to severe damages to the user, as well as inefficacy of medications in many situations. The pharmacist then plays a key role in guiding the use of these products and in administering them when necessary.

Also in this matter of clinical services, in Brazil, the pharmacist can provide a prescription of medicines exempt from medical prescription, herbal medicines and food supplements, within their professional and legal scope (RDC CFF 586/13 and RDC CFF 661/18), when the need for these is verified, always taking into consideration the actions of the other professionals involved with the athletes. This prescription can be of great value to assist in moments of therapy or even for questions where the pharmacist detects that these products can bring some benefit to the athlete [7,53].

In summary, the participation of the pharmacist as a member of the multidisciplinary team related to sports, in the most diverse modalities, can be of great value, since the use of medicines is often necessary. The integration of this professional into teams of collective modalities such as soccer, volleyball, and basketball, as well as individual ones, can complement the actions of physiologists, nutritionists and doctors, in a synergism that tends to optimize the use of substances when necessary, and prevent possible damaging events for athletes.

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
