# Peer review of "Pharmaceutical Care in Sports"

_pharmacy, 2020, doi:10.3390/pharmacy8040218_

Round 1

Reviewer 1 Report

I read this article with great interest.  The topic is one that surely needs some more exploration. The authors did a wonderful job of laying the groundwork of the literature to date.  As this is a review article, the two aims the authors set out to do were: "the aim of the present work was to describe the role of the clinical pharmacist and its actions related to the sport practice, in the context of pharmaceutical care, and demonstrate how this specific service can optimize the use of drugs, supplements and herbal medicines by athletes and physical activity practitioners"

Revisions

  1. A systematic way was not provided by the authors on the method of how the review was performed. Would it be possible to state how the review was performed?
  2. The review certainly described the role of a clinical pharmacist, however, I think Line 76-79 can be further elaborated on to help build the framework. Would it be possible to clarify and elaborate on this section?
  3. The second aim to demonstrate specific services may need to also be more elaborated on.  I see that using a SOAP note or TOM or WEIGHT$ can be the formal structure, but the authors should elaborate further on it.  Line 238-245 explains the WEIGHT$ structure, but for some one unfamiliar with the acronym explain it first may be helpful. Would it be possible to clarify and elaborate on this section?

Author Response

Dear reviewer

Reviewer 2 Report

Abstract:

Line 9….line 14:   With the sentence starting with “Once a range…….when it’s necessary”.  The purpose of having a clinical pharmacist within the multi professional team is realized within this sentence.  This is a long, run-on sentence that needs to be broken up into 2-3 sentences. I believe the content and thought process is very good discussing the historical and current importance of the services the clinical pharmacist can provide, each needs to be broken up into a different sentence. 

Introduction:

Line 21:   “recovering” , would use “recovery” instead

Line 22:  “proposes” should be “purposes”

Line 29: “ is used to define a broad set of clinical actions”

Line 36…Line 42:  consider breaking this thought into 2-3 sentences if possible.

Line 48:  the reference at the end of the sentence should be (4-7)  not (4,7)

Line 57: “directed to patients” should be “directed from patients”

The pharmacist and the sports practice:

Line 63  “relation” to  “relationship”

Line 79 reference listed should be (9-12)

Line 89:  “are still little explored in scientific research” – needs to be re-worded

Dietary supplements, herbals and drug use by athletes

Line 101: “practitioners initiate  the activities”   there is a large space between initiate and the, formatting issue

Line 116:  reference should be listed as (18-21)

Line 127:  change phrasing to “and other various syndromes (ie Turner,Cushing) and androgen replacement.

Line 128:  reference (22-25)

Line 129:  change to “ are the testosterone esters:  oxandrolone, nandrolone…etc”

Line 136: reference (26-29)

Line 137: athletes omit what fact?  Sentence incomplete

Line 138: sentence starting with “Few……”  sentence seems very confusing, not sure what information they are passing along.  Sentence needs to be re-written

Line 159: reference (35-37)

Line 172:  sentence ending with composition.  Need to rephrase the next sentence to start the next thought.  That sentence is difficult to follow.

Line 192:  “improves” should be “improve”

Line 203:  “or in estrogen” should be “or into estrogen”

Line 204: consider changing sentence to among adverse reactions include: alterations to lipid profile, hepatic and renal function and hematocrit.  Then start the next sentence then describe the next set of side effects including gynecomastia etc

Line 212: “that” should be “those”

Line 228:  rephrase  “allowing to find possible errors”

Line 231:  please put a reference for where to locate dader method, pharmacotherapy workup and therapeutic outcomes monitoring (TOM) to easily locate for the reader

Line 239  “example of a questionnaire”

Line 242: do not need to start a new paragraph if you are referring to the protocol used be Morris (2015)

Line 242:  the first sentence…”in this protocol, are evaluated the purpose of the use of the supplement”, this is not a complete sentence, not sure what in the protocol is being referred to

Line 256: do not start a new paragraph is still referring to the article by Howard et al. put the reference at the end of that paragraph

Line 261 use “considered” not “considerate”

Line 264: “by a pharmacist”

Line 267:  “and a lot of publicity given to these products can lead the general public”

Line 272 “There are a vast amount of literature that describe how…..”

Line 287: “the pharmacist can provide a prescription…..”

Author Response

Dear reviewer
